# The Impact Mechanism of Digital Transformation on the Risk-Taking Level of Chinese Listed Companies

**Debao Dai [1], Shengnan Han [1,\*], Min Zhao [2] and Jiaping Xie [3]**

1   School of Management, Shanghai University, Shanghai 200444, China
2   SILC Business School, Shanghai University, Shanghai 201800, China
3   School of Business, Shanghai University of Finance and Economics, Shanghai 200433, China
\*   Correspondence: han980216@shu.edu.cn

**Abstract:** As the core engine of the digital economy, the digital transformation can make modern enterprises survive and develop better now. By the sample data of listed companies in the years from 2015 to 2020, this paper identifies the degree of enterprise digital transformation through text analysis, empirically examines the impact mechanism of digital transformation on corporate risk-taking, and fully considers the heterogeneity problems. The findings are as follows: (1) Digital transformation can improve the level of enterprise risk taking, especially the improvement of enterprise financial stability and strategic risk taking; (2) in terms of enterprise attribute structure, digital transformation can significantly enhance the risk-taking level of non-state-owned enterprises and high-tech enterprises; (3) the mechanism identification test finds that innovation-driven and enterprise value enhancement play a strengthening role in the role of digital transformation in promoting enterprise risk-taking level, and resource allocation efficiency as a mediating path weakens the role of digital transformation on enterprise risk-taking level. This study provides a basis for promoting the improvement of enterprises risk-taking: digital transformation can help enterprises maintain financial stability, improve innovation output capacity, enterprise value level, enterprise risk-taking capacity and sustainable development. At the same time, the Chinese government should take measures to further stimulate the willingness of state-owned enterprises to digital transformation.

**Keywords:** digital transformation; risk-taking; innovation-driven; value enhancement; resource allocation efficiency

## 1. Introduction

The digital economy is the future development direction of the world. Under the guidance of the "the 14th Five-Year Plan for China's National Economic and Social Development", the State Council issued the "14th Five-Year Plan for Digital Economy Development" to promote the deep integration of traditional economy and digital economy, and thus achieve digital transformation. As an important focus of the development of the digital economy, digital transformation has empowered all industries and fields, and become the core driving force to stimulate the innovation vitality of enterprises and promote the high-quality transformation of the economy (White Paper on Digital Transformation of Enterprises, 2021). The Corona Virus Disease 2019 has inspired the digital transformation of Chinese enterprises to break out at both the demand side and the supply side, and the digital transformation of traditional enterprises has changed from the "optional" of industry-leading enterprises to the "mandatory" of most enterprises. However, some studies still have a lot of questions about the potential value of digital transformation, arguing that not all businesses can benefit from it [1]. According to the "2021 China Enterprise Digital Transformation Index" research report released by Accenture, considering the epidemic situation and changes in international relations, 80% of enterprises have tried to carry out digital transformation, but only 16% of enterprises have achieved significant

digital transformation results, truly releasing digital potential, while other enterprises may be due to lack of talent, weak transformation foundation, or weak "hematopoietic" function of enterprises, coupled with the further lag of external "blood transfusion" mechanism, making enterprises invest a lot of money in digital transformation, but revenue did not increase, resulting in business difficulties, and if serious, may lead to enterprise bankruptcy. On the contrary, on the positive side, there are huge opportunities in the risk. The willingness of enterprise digital transformation has increased due to the impact of the epidemic, and with higher digital maturity, the revenue growth rate of the leading enterprises in 2020 is 3.7 times that of other enterprises. The core competitiveness of enterprises has been consolidated, and the value of enterprises has gradually improved, thus enterprises have a higher level of risk-taking, and at the same time, enterprises are willing to bear greater risks in order to obtain greater benefits.

Risk-taking is an important part of corporate strategic decision-making, reflecting the risk appetite of enterprises when considering investment projects [2]. Enterprises, as economic subjects, play a prominent role in promoting economic and social development. Meanwhile, Risk selection is the key to enhancing the core competitiveness of enterprises and promoting high-quality economic development. The increase in the risk-taking level and the pursuit of excessive profits are also the driving force for sustainable economic growth [3]. Therefore, a reasonable level of corporate risk-taking can help enterprises make decisions to take risks and seize investment opportunities in order to obtain excess profits, promoting the improvement of corporate performance [4] and enhancing the core competitiveness of enterprises. However, due to the uncertainty of the environment and the opacity of information, managers often tend to choose conservative investment projects under the guidance of risk aversion motives, so that enterprises are at a lower level of risk taking, which in the long run will reduce the survival motivation and vitality of enterprises, and thus be eliminated by the market. The application of digital technology can promote the improvement of enterprise innovation level [5], make full use of enterprise resources, enhance enterprise value [6], and increase the willingness of enterprises to take risks in investment activities. On the other hand, digital transformation uses artificial intelligence, cloud computing, blockchain, and other digital technologies to break the limitations of time and space through data integration and all-round changes to economic factors, to monitor and warn production and operation around the clock, and to create a favorable environment for the improvement of enterprise risk-taking level. Therefore, under the trend of the digital economy era, it is of great theoretical and practical value to explore the influence mechanism between digital transformation and corporate risk-taking.

Based on the above analysis, this paper takes Chinese listed companies from 2015 to 2020 as research samples, uses Python to identify the degree of digital transformation and uses fixed effect model to empirically test the impact mechanism of digital transformation on enterprise risk bearing, and strives to open the black box of the impact mechanism between digital transformation and enterprise risk taking from three aspects: innovation input and output, resource allocation efficiency, and enterprise value. The possible marginal contribution of this paper is that: from a research philosophy perspective, most literature researches on digitalization and corporate risk-taking level mainly consider the external macro environment [7,8]. Based on the micro level of enterprises, this paper links the degree of enterprise digitalization transformation with the corporate risk-taking level, and analyzes the relationship between "enterprise digitalization transformation–innovation input and output, resource allocation efficiency and enterprise value–corporate risk-taking level", which enriches the economic consequences of digital transformation, deeply understands the level of enterprise risk bearing from the perspective of risk prevention and control, and unveils the mechanism "black box" between digital transformation and corporate risk-taking; From the research content, this paper highlights the impact of digital transformation as an emerging influencing factor on the financial stability and strategic risk-taking level of enterprises, and further demonstrates the potential value of digital transformation on the risk-taking level of enterprises; In terms of research results, this paper demonstrates

that the role of digital transformation in improving the level of enterprise risk taking is different in terms of enterprise characteristics, and the impact on private enterprises and high-tech enterprises is more significant. This discovery helps the Chinese government to take precise policy guidance based on the differences of enterprise characteristics.

## 2. Literature Review

### 2.1. Research on the Necessity and Promotion Mechanism of Digital Transformation

Under high-quality economic development, digital transformation has become a hot topic in academic research. However, the quantitative measurement method of enterprise digital transformation has not yet been unified by the academic community. During the early rise of digitalization, considering that the digital transformation of enterprises is a systematic project reflected in the overall operation of enterprises, in order to comprehensively examine the degree of digital transformation of enterprises, some scholars have created a digital maturity assessment model by using a scale, and digital maturity is also recognized as a standard for digital transformation by academia and industry [9]. However, the dimensions adopted are different. Some scholars built digital maturity models based on four dimensions: culture, organization, insight and technology [10]; while others measured the digital maturity of enterprises in terms of three dimensions: IT technology, the use of digital technology to support business activities, business information communication and business process integration [11]. Considering that it took a long time to collect data using the scale and the data collected were subjective, scholars argued that for listed companies, the use of words in the annual report can reflect the current status and future strategic direction of the company, so it is proposed to use the number of words related to digitalization in annual reports to measure digital transformation indicators [12]. Meanwhile, some scholars argued that measuring the degree of digitalization of enterprises by the frequency of keywords in the annual report lacks professional subjective judgment and is prone to misjudgment, so they set up a dumb variable combining text analysis and manual judgment to investigate the impact of digital transformation on audit quality [13].

For the research related to enterprise digital transformation, the most easily thought of research direction is the positive promotion mechanism of enterprise digital transformation on economic consequences, and many scholars have studied from multiple dimensions. The economic consequences of enterprise digital transformation mainly reflect organizational flexibility, customer value, technology market, innovation performance, and capital market. Some scholars argued that digital transformation can help to improve enterprise innovation performance [14] by facilitating innovation, absorption, and adaptation of enterprises [15], improve organizational resilience [16], and promote the expansion of a country's technology market [17]. As for SMEs, digital transformation contributed to the innovation of business models, creating new distribution channels and new ways to provide and create value for the customer base [18]. In addition, from the perspective of capital markets, digital transformation has significantly improved equity liquidity, providing clues for understanding the liquidity of micro-entities in the capital market [12]. Although the advantages of digital transformation and advanced digital technology are obvious, risks are often hidden, and the risks arising from digital transformation need to be supplemented, but there is little literature to examine the impact of digital transformation from a risk prevention perspective.

### 2.2. Research on the Influencing Factors of Enterprise Risk-Taking

Based on the previous literature, it is found that the connotation of the concept of enterprise risk-taking is mainly reflected in three aspects: the willingness to take risks, the level of risk-taking, and the ability to bear risks [19]. In previous studies, there is relatively little literature that makes a strict distinction between the three concepts and more literature on the influencing factors of corporate risk-taking. Among the available studies, most of the factors influencing corporate risk-taking have been discussed in terms of the internal characteristics and external environment characteristics of the enterprise. At

the micro level, scholars analyze the influencing factors of corporate risk-taking based on the perspectives of and the gender ratio of management teams [20], managers' risk appetite [7], performance evaluation mechanisms [21], knowledge management capabilities [22], and the nature of enterprises [23], venture capital [24]. At the macro level, studies have found that global financial conditions [25], the level of digital economy development [8], and government loan guarantee support [26] have a positive impact on corporate risk-taking, while economic policy uncertainty [27], customer concentration [28] have greatly reduced the corporate risk-taking.

### 2.3. Digital Transformation and Enterprise Risk Taking

Currently, there is less research literature directly related to enterprise digital transformation and enterprise risk taking, but the path of influence between the two can be inferred from other similar literature. Some scholars used semantic analysis to reconstruct the definition of digital transformation from an existing definition, namely "aimed at improving the process of an entity by triggering a significant change in the attributes of an entity through a combination of information, computing, communication, and connectivity technologies" [29]. In terms of the macro environment, Some studies constructed a provincial digital economy development index from the level of digital industrialization and industrial digitalization, and found that the digital economy pulled the level of innovation of regions and the financing ability of enterprises, thus contributing to the level of enterprise risk-taking [8]. Meanwhile, in the era of vigorous development of the digital economy, other scholars further refined the research level of the digital economy. They found that the breadth of coverage and depth of use of regional financial technology also had a positive impact on the level of enterprise risk-taking [30], and the development of digital economy and financial technology provided the external impetus for enterprise digital transformation—support of technical and financial conditions, laying a certain foundation for the successful digital transformation of enterprises. From the micro level of enterprises, some scholars analyzed the case of Renhe Group and found that the digital service transformation of manufacturing industry can evolve into three models: online retail, bilateral platform and ecological network, thus transforming the way of value acquisition and creating new value [31]. Some scholars used a spiral model to reveal the transformation of new technologies on enterprise structure, innovation, and performance in the context of digital transformation [32] and thus use emerging technologies to improve risk control [33]. From this perspective, digital transformation empowered enterprises with greater economic vitality and organizational resilience as well as business potential enhanced firms' willingness to take risks in decision-making and the overall risk-taking capacity of the organization, and ultimately creates positive feedback on enterprise risk-taking. Given this, this paper intends to identify and test the impact and mechanism of "digital transformation of enterprises to enterprise risk-taking", as well as the heterogeneous effects under different scenarios, so as to provide new evidence for understanding digital transformation and corporate risk-taking in Chinese listed companies.

The rest of this study is organized as follows: the third part analyzes the impact path of digital transformation on the risk-taking level of enterprises and proposes hypotheses; the fourth part constructs an econometric model and gives explanations of the relevant variables and the sources of variables; the fifth part analyzes and discusses the empirical results and conducts stability and heterogeneity tests; the sixth part tests the impact path of digital transformation on the risk-taking level of enterprises; the seventh part is a summary of the article and makes reasonable recommendations.

## 3. Theoretical Analysis and Hypothesis Formulation

In the wave of digitalization, digital transformation provides an effective boost for enterprises to seize the commanding heights of new competition, and gives enterprises a new life to survive in the new era. For traditional enterprises, digital transformation is not a multiple-choice question, but a survival question. On the one hand, the digital

transformation of enterprises will trigger financial turmoil and put them in operational difficulties. As an enterprise development strategy, digital transformation has high risk and high uncertainty, and digital transformation as "all-scene", "all-connected" and "all-intelligent" transformation requires more investment in digital technology and staff skill development, which requires continuous capital investment, and these capital investments and the waste of time costs in the process bring certain financial risks to enterprises. In addition, under the impact of the tide of the digital economy and the epidemic, some enterprises blindly carry out digital transformation in order to break through the business difficulties, ignoring the company's informatization, digital core technology, and managers' clear understanding of the digital transformation process, which leads to encountering the risk of not turning and cannot turn in the digital transformation, and making enterprises into a worse business dilemma. Based on this consideration, in order to avoid enterprises falling into trouble, enterprises tend to be less willing to take risks.

**Hypothesis 1.** *Digital transformation will reduce the risk-taking level of enterprises, financial risk-taking level and strategic risk-taking level.*

However, on the other hand, digital technology responds to complex environmental changes inside and outside the enterprise, improves the availability of information, promotes the sharing and accurate allocation of resources among organizations within the enterprise, and expands innovation opportunities, thereby maximizing the benefits of the enterprise. In addition, digital technology monitors and warns the whole process of production and operation throughout the day, helping enterprises to prevent in advance and propose solutions to the dilemmas they will face. From this perspective, the change in the digital transformation of enterprises is precisely conducive to reducing financial risk and improving the level of risk-taking of enterprises, that is, enterprises are willing to take a higher degree of risk for this purpose in pursuit of higher returns [34]. Based on this, this paper will explore the impact of digital transformation on the risk-taking level of enterprises and its mechanism from three main paths: innovation-driven, resource allocation efficiency, and enterprise value. As shown in Figure 1.

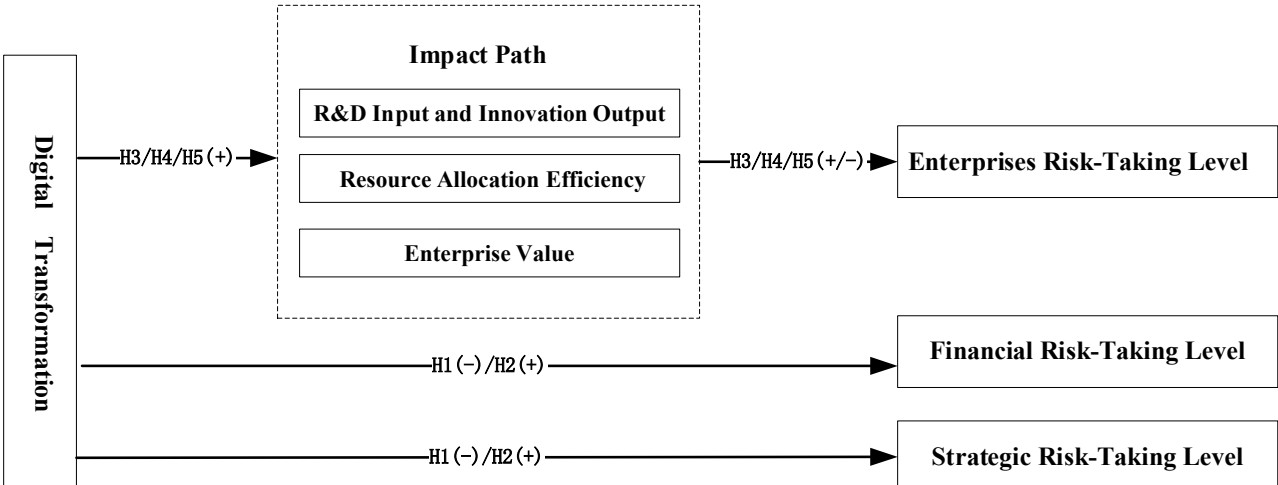

**Figure 1.** Mechanism of the effect of digital transformation on enterprises risk-taking level. Note: "+" indicates a positive promoting effect, "−" indicates a negative inhibiting effect.

**Hypothesis 2.** *Digital transformation can significantly improve the level of enterprise risk-taking, financial risk-taking and strategic risk-taking.*

The digital transformation of enterprises can strengthen the innovation momentum, thus improving the level of enterprise risk-taking. Firstly, at the enterprise level, digital transformation is the core development strategy of enterprises, and in order to achieve this

strategic goal, enterprises will provide "digital" assistance in the whole system and in all aspects; At the market level, under the requirements of high-quality economic development, the construction of digital economy has become a national policy, enterprises with a high degree of digitalization are more likely to be accepted by the market, and enterprises often have a stronger incentive to increase the intensity of research and development and further promote the digital transformation of enterprises. Secondly, the introduction of digital transformation tools represented by cloud computing and big data in the context of the digital era not only improves the efficiency of internal data processing, but also accelerates the response speed of enterprises to changes in the external environment and expands the innovation opportunities of enterprises. Meanwhile, the process of digital technology to promote innovation has shifted from extensive to precise, improving the innovation efficiency of enterprises [15]. Furthermore, when the degree of digitalization of enterprises reaches a certain level, enterprises can digitize the innovation research process by introducing digital technology, which greatly reduces R&D investment and achieves innovation goals at a smaller cost and faster speed [35], and motivates enterprises to bring more innovative output effects and increases their core competitiveness, thus promoting a positive feedback mechanism of innovation momentum on the level of enterprise risk-taking.

**Hypothesis 3.** *Digital transformation of enterprises can strengthen innovation dynamics and thus increase risk-taking levels.*

In the context of the digital economy, the living environment faced by enterprises is more severe. To improve sustainable development ability and long-term competitiveness, it is undoubtedly a "favorable" choice to promote the improvement of limited resource allocation efficiency. Through the application of digital technologies, enterprises can obtain a large amount of information from the whole chain of "production–supply–marketing" and analyze it to effectively reduce the information asymmetry between the upstream and downstream of the industrial chain and achieve the optimal allocation of resources in the entire supply chain [36]. The improvement of resource allocation efficiency can effectively reduce the production cost and management cost of enterprises, thus avoiding the financial risk of enterprises. In addition, through digital transformation, enterprises break down business boundaries and organizational boundaries of enterprises and build flexible organizational forms. At the same time, the development of enterprise intelligence replaces the low-end labor force, improves the skill demand of the labor force, and optimizes the human capital structure of enterprises. With the improvement of human capital and high-quality knowledge capital, enterprises can quickly adapt to changes in the enterprise environment through self-learning and inter-group learning, thereby optimizing the factor resources they possess and reducing the strategic risk of the enterprise. Therefore, the digital transformation of enterprises promotes the improvement of resource allocation efficiency, and the optimization of enterprise's factor allocation disperses risks to a certain extent, so that the risks faced by enterprises are not huge, and so enterprises do not need to urgently improve their risk management capabilities.

**Hypothesis 4.** *The digital transformation can improve the efficiency of resource allocation, but the efficiency of resource allocation transmits the inhibitory effect of digital transformation on risk taking.*

Digital transformation of enterprises can effectively increase enterprise value and thus improve enterprise risk-taking. In essence, an enterprise is a system for creating, transmitting and acquiring value. In the era of material economy, the way for enterprises to obtain value is to achieve large-scale development efficiency based on the specialized division of labor, so as to obtain long-term returns, and the development of intelligence has profoundly changed the value model of enterprises. Digital transformation promotes the nature of all-round changes in enterprises, and systematically empowers all aspects of enterprise operation and management, such as production and operation optimization,

product or service innovation, and format transformation, so that the intrinsic value of enterprises is continuously improved [5] and the anti-risk ability of enterprises is enhanced. What is more, enterprises with better digital transformation can convey information about their internal operation status to the outside world by taking advantage of their digital advantages, thus attracting a large number of investors and obtaining sufficient resources investment, which in turn improves their risk-taking ability.

**Hypothesis 5.** *Digital transformation of enterprises can improve the level of enterprise risk-taking through value-enhancing.*

## 4. Research Design

### 4.1. Data Resources

Given that the government issued a number of Internet technology-led industrial policy programs for the digital transformation of the real economy between 2015 and 2016, which empowered enterprises to actively promote digital transformation, this paper takes 2015 as the initial year and selects the data of A-share listed companies in Shanghai and Shenzhen from 2015 to 2020 as the research sample. The data is processed as follows: (1) excluding enterprises with missing major variables; (2) excluding the stocks in ST, * ST or PT status, such stocks represent that the enterprise is currently in an abnormal operating state, that is, it is in a loss state for more than two consecutive years, and there is a risk of delisting (3) eliminating financial listed enterprises according to 2012 SEC industry classification standards. Finally, 12216 observations containing 2837 companies are obtained, which are unbalanced panel data. Meanwhile, to avoid the influence of abnormal values, all continuous variables are Winsorize shrunken at the upper and lower 1% level in this paper. The annual reports of listed companies are obtained from Juchao Information Website, other financial data are obtained from Cathay Capital Database (CSMAR) and China Research Data Service Platform (CNRDS), and the GDP of each province in the control variables are obtained from China Statistical Yearbook.

### 4.2. Variable Description

#### 4.2.1. Explained Variable

Corporate Risk-taking (risk). Drawing on the practices of Habib and Hasan (2017) [37] and Wang Xiuli (2022) [38], this paper uses the return on assets (roa) to represent corporate profitability, expressed as the ratio of profit before interest and tax to total assets. Considering the impact of industry and cycle, the enterprise's return on assets (roa) is subtracted from the annual industry average to obtain the industry-adjusted enterprise's total return on assets(adj_roa). In this study, the standard deviation of the industry-adjusted enterprise total return on assets (adj_roa) within three years ($t - 2$ to $t$ years) during the observation period is used to measure the enterprise risk-taking level (risk1), and the range of the industry adjusted enterprise total return on assets (adj_roa) during the observation period is used as the robustness test (risk2). The specific calculation method is shown in Equations (1)–(3):

$$risk1_{i,t} = \sqrt{\frac{1}{T-1} \sum_{t=1}^{T} \left( \text{adj\_}roa_{i,t} - \frac{1}{T} \sum_{t=1}^{T} \text{adj\_}roa_{i,t} \right)^2}, T = 3 \tag{1}$$

$$risk2_{i,t} = Max(\text{adj}_{roai,t}, \text{adj}_{roai,t-1}, \text{adj}_{roai,t-T+1})$$
$$- Min(adj_{roai,t}, adj_{roai,t-1}, adj_{roai,t-T+1}), \ T = 3 \tag{2}$$

$$adj_{roa} = \frac{EBIT_{i,t}}{ASSET_{i,t}} - \frac{1}{X} \sum_{k=1}^{X} \frac{EBIT_{k,t}}{ASSET_{k,t}} \tag{3}$$

Additionally, this study further describes in detail the impact of digital transformation on different levels of risk-taking. Considering financial risk as the main disclosure of risk exposure and the fact that financial stability to some extent represents an indicator of how much risk a company is willing to take in order to gain excess profits. Strategic risk is an important risk indicator that determines whether a firm can survive in the long run

and gain profits in the future as well as how a firm chooses the level of risk-taking in a complex internal and external environment, which is also an important strategic decision-making issue [39]. In this study, the financial risk-taking level (Zscore) and the strategic risk-taking level (Strategy) are chosen to investigate the impact on different risk-taking levels during the digital transformation process, respectively. The financial risk-taking level is represented by the Z-score, which is calculated by equation (4); a larger Z-score indicates a higher financial stability and a higher level of financial risk-taking. Strategic risk-taking level (Strategy) is measured by factor analysis of the three variables of R&D expenditure, capital expenditure, and long-term debt to form a composite indicator [40]. The results of the factor analysis showed that the KMO was 0.715 and the Bartlett's spherical test reached a significant level, and the cumulative percentage of the factor analysis was 74.52%, indicating that R&D expenditures, capital expenditures, and long-term liabilities can be better aggregated into one indicator.

$$Z = 1.2X_1 + 1.4X_2 + 3.3X_3 + 0.6X_4 + 0.999X_5 \tag{4}$$

where $X_1$ = working capital/total assets, the value of which comprehensively reflects the firm's liquidity and size characteristics; $X_2$= Retained earnings/total assets, which reflect the long-term profitability of the enterprise; $X_3$= EBIT/total assets, which reflects the profitability of the company's assets; $X_4$ = market value of equity/book value of total liabilities, which reflects the solvency of the enterprise; $X_5$ = operating income/total assets, which measures how efficiently the firm utilizes its assets.

### 4.2.2. Explanatory Variable

Enterprise Digital Transformation (Infrequent). In the new era, the digital transformation of enterprises is a major strategy for the high-quality development of enterprises, so the degree of use of digital technologies can be reflected from the frequency of the keywords involved in the strategy in the periodic report. The number of keywords indicates the strategic characteristics and development prospects of the enterprise, and largely reflects the intensity of the company's digital transformation [13]. Drawing on previous studies [41], the paper uses Python to crawl the annual reports of all listed companies on CNINFO, convert them into txt files, and then extracts all the text contents through the pdf2txt library. In order to facilitate the statistics of the frequency of keywords, the data of company name, shareholders and other basic information of the company are excluded from the stop-word. Referring to the research of Wu F et al. (2021), this paper summarizes 84 keywords related to it from five aspects: artificial intelligence, blockchain, cloud computing, big data, and the application of digital technology [12], and takes the sum of the frequency of keywords in the annual report as the proxy indicator of the degree of digital transformation.

Figure 2 depicts the trend of the degree of digital transformation in different industries from 2015 to 2020. On the whole, the digitalization of all industries is accelerating, among which the digitalization of information transmission, software information services and leasing and business services is relatively high. Influenced by the development of the Internet and the epidemic, online classes and online teaching are developing at a high speed, so the education industry is undergoing a faster digital transformation. In contrast, the degree of digital transformation in mining and power, heat production and supply has been at a low level.

### 4.2.3. Control Variables

To ensure the validity of the model and get accurate regression results, this paper refers to the existing research [12,42], and adds a series of control variables from the three aspects of company characteristics, internal control and external environment. Company characteristic variables include the age of the listed company (Intime), asset-liability ratio (lev), enterprise growth (grown), intangible asset ratio (IA), and mobility ratio (crate); The management level of internal control considerations mainly includes ownership concentration (first), CEO duality (dul), board size (Inboard), and management shareholding ratio

(mrate); At the same time, the gross domestic product (Ingdp) is used as a control variable of the external environment that affects the level of risk-taking of enterprises. The variables used in this paper and their definitions are shown in Table 1.

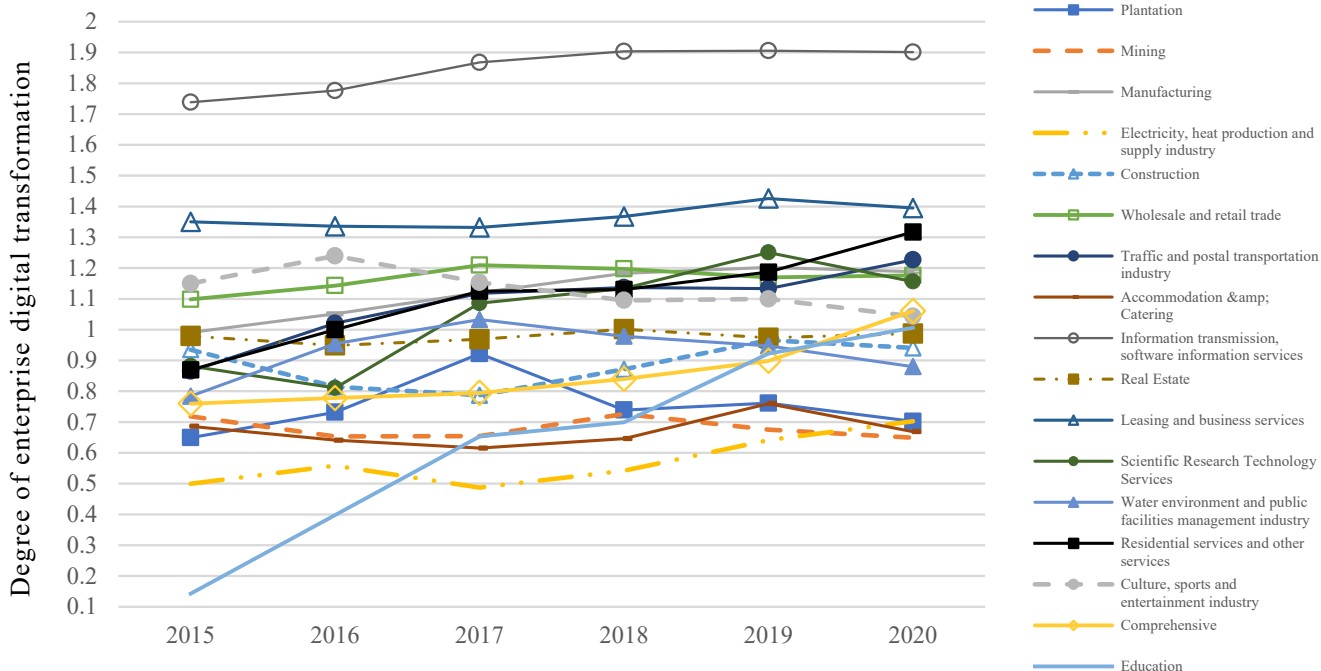

**Figure 2.** Trends in the degree of digital transformation in different industries. Data source: Python crawling, author sorting.

**Table 1.** Variable descriptions.

| Variables | Brief | Description |
|---|---|---|
| Enterprise risk-taking level | risk1 | Fluctuation of return on enterprise assets |
| Financial Risk-taking level | Z-score | Z-value |
| Strategic Risks-taking level | Strategy | Factor analysis |
| Digital transformation | Infrequent | Logarithm of keyword frequency in the annual report |
| Enterprise growth | grown | Enterprise revenue growth rate |
| Age of the listed company | Intime | The logarithm of (time to measurement minustime to market). |
| Asset liability ratio | lev | Total company liabilities divided by total assets |
| Intangible assets ratio | IA | Net intangible assets/total assets |
| Mobility | crate | Current Ratio |
| Ownership concentration | first | Ratio of the number of shares held by the largest shareholder to the total number of shares |
| Management shareholding ratio | mrate | Number of shares held by management/total number of shares |
| CEO duality | dul | Whether the two powers are separated, the chairman and the general manager are the same person is recorded as 1, otherwise is 0 |
| Board Size | lnboard | Logarithm of number of directors |
| Gross Domestic Product | lngdp | GDP in logarithm |

### 4.3. Model Construction

To investigate the impact mechanism between digital transformation and enterprise risk taking, this paper establishes an econometric model as shown in Equation (5):

$$risk_{i,t} = \alpha + \beta Infrequent_{i,t} + \sum \varphi CVs_{i,t} + \sum year + \sum Indenty + \varepsilon_{i,t} \tag{5}$$

where $i$ denotes the firm, t denotes time, and $risk_{i,t}$ denotes the explained variable, that is, the level of enterprise risk-taking, and $Infrequent_{i,t}$ denotes the explanatory variable, that

is, the degree of digital transformation of the firm, and $CVs_{i,t}$ denotes the set of control variables mentioned above, and $\varepsilon_{i,t}$ denotes the random error term. In consideration of the reliability of the regression results, this paper also made the following treatment: (1) This paper simultaneously controlled the dummy variables of time and Industry to absorb fixed effects as much as possible, where the industry variables are classified with reference to the 2012 industry classification standard of the SEC. (2) Cluster clustering robust standard errors are used to adjust the t-statistics in all back equations.

## 5. Empirical Results and Economic Explanation

### 5.1. The Influence of the Digital Transformation on Corporate Risk-Taking

Table 2 shows the descriptive statistics of the main variables for the years 2015–2020. From the descriptive statistics: first, the average value of risk1 is 0.0321, and the value range of risk1 is −0.799 to 0.459, which indicates that there are differences in risk-taking levels among different enterprises. risk2 also has the same trend. Secondly, the average value of lnfrequent is 1.948, and the value range of lnfrequent is 0 to 6.250, which indicates that the degree of digital transformation of enterprises in China is not high, and there are large differences between enterprises. Third, the average value of the financial risk-taking level (Zscore) is 5.288, the minimum value is −25.80, and the negative symbol indicates to a certain extent that some enterprises still have large financial risks, while the maximum value is 419.8, indicating that the company's operating conditions have maintained the best state, and the company has maintained a profitable state during this period. Compared with this, there are small differences in the enterprise's strategic risk-taking level.

**Table 2.** Descriptive statistics for variables.

| Variable | Obs | Mean | Std. Dev. | Min | Max |
|---|---|---|---|---|---|
| risk1 | 12,216 | 0.0321 | 0.0503 | −0.799 | 0.459 |
| risk2 | 12,216 | 0.0605 | 0.0927 | −1.460 | 0.906 |
| Zscore | 12,216 | 5.288 | 9.990 | −25.80 | 419.8 |
| Strategy | 11,861 | 0.0121 | 1.118 | −33.01 | 35.08 |
| lnfrequent | 12,216 | 1.948 | 1.383 | 0 | 6.250 |
| first | 12,216 | 0.336 | 0.147 | 0.0287 | 0.990 |
| lnboard | 12,214 | 2.111 | 0.199 | 1.386 | 2.833 |
| mrate | 12,216 | 14.36 | 19.79 | 0 | 98 |
| lev | 12,216 | 0.428 | 0.214 | 0.0197 | 4.995 |
| dul | 12,216 | 0.301 | 0.459 | 0 | 1 |
| lntime | 11,682 | 2.099 | 0.868 | 0 | 3.912 |
| grown | 12,216 | 0.342 | 5.110 | −1.309 | 429.0 |
| crate | 12,216 | 2.315 | 2.370 | 0.0278 | 49.63 |
| lngdp | 12,216 | 10.68 | 0.692 | 6.950 | 11.62 |

When selecting the control variables, in order to prevent endogeneity problems caused by omitted variables, this research puts all the basic information such as individual firm characteristics and internal management characteristics into the variables for control as much as possible, but in fact, there may be strong correlations between these variables, such as the correlation between the time to market and the growth of the company, and Zscore also includes the profitability and asset liquidity ratio of the company. The mutual influence of variables can cause more serious multiple co-linearity problems and make the accuracy decrease, so we use the variance inflation factor method (VIF) to test the multiple co-linearity problems between variables, and the results obtained are shown in Table 3. The results indicate that the VIF values of each variable are less than 10 and the mean value of VIF is 1.39, which is less than 2, indicating that the multicollinearity problem is within the normal range, and accurate regression results can be obtained for these variables.

**Table 3.** Multicollinearity test.

| Variable | VIF | 1/VIF |
|---|---|---|
| lev | 2.470 | 0.404 |
| Zscore | 2.450 | 0.409 |
| crate | 1.670 | 0.598 |
| MRat | 1.550 | 0.646 |
| lntime | 1.540 | 0.647 |
| dul | 1.110 | 0.904 |
| lnboard | 1.090 | 0.921 |
| lngdp | 1.070 | 0.937 |
| first | 1.060 | 0.939 |
| lnfrequent | 1.060 | 0.947 |
| IA | 1.030 | 0.967 |
| Strategy | 1.030 | 0.971 |
| grown | 1.000 | 0.998 |
| Mean VIF | 1.390 | |

Considering that the rationality of model selection seriously affects the accuracy of the study results, this study first used F test to determine which regression effect of fixed effect and mixed OLS regression was better. The test results showed that Prob > F = 0.000, so the original hypothesis was rejected, indicating that the fixed effect was more appropriate. Secondly, this study conducted Hausman test to select the random effect model and fixed effect model, and the results of the study showed that the $P$-value was 0.000 < 0.05, indicating that the original hypothesis was rejected and the fixed effect results were better. Combining the F-test and Hausman test, the fixed effect model was finally selected.

Table 4 reports the full sample regression results of digital transformation and enterprise risk taking level. Column (1) reports the regression results of digital transformation to enterprise risk-taking. The results of the study show that the regression coefficient of lnfrequent and risk1 is 0.001, the $t$-value is 2.20, and it is positively significant at the 5% level, which indicates that the digital transformation promotes the increase in the level of corporate risk-taking, that is, the deeper the degree of digital transformation, the greater the willingness of enterprises to take risks in order to obtain greater benefits. Moreover, the use of digital technology makes the profitability of enterprises improve rapidly, and the comprehensive strength of enterprises is continuously strengthened, so the higher the level of corporate risk-taking. The reason is that enterprises use digital technology to carry out digital transformation, which improves the innovation efficiency of enterprises, optimizes the resource allocation of enterprises, promotes the promotion of enterprise value, and thus enhances the anti-risk level of enterprises. Therefore, Hypothesis 2 of this paper is supported by empirical evidence.

**Table 4.** Results of the full sample regression.

| | (1) | (2) | (3) |
|---|---|---|---|
| VARIABLES | risk1 | Z-Score | Strategy |
| lnfrequent | 0.001 ** | 0.010 ** | 0.011 *** |
| | (2.20) | (2.56) | (3.65) |
| first | −0.029 *** | 0.129 *** | 0.220 *** |
| | (−12.60) | (3.75) | (7.19) |
| lnboard | −0.013 *** | −0.171 *** | 0.115 *** |
| | (−5.83) | (−5.25) | (4.10) |
| dul | 0.003 *** | 0.003 | −0.000 |
| | (3.34) | (0.24) | (−0.06) |
| mrate | −0.000 | −0.002 *** | −0.001 *** |
| | (−0.61) | (−7.47) | (−5.03) |
| lev | 0.008 *** | −2.574 *** | 0.395 *** |
| | (2.79) | (−60.10) | (12.35) |

**Table 4.** *Cont.*

|  | (1) | (2) | (3) |
|---|---|---|---|
| **VARIABLES** | **risk1** | **Z-Score** | **Strategy** |
| grown | −0.006 *** | 0.121 *** | 0.005 |
|  | (−4.49) | (7.16) | (0.56) |
| crate | 0.000 | 0.116 *** | 0.012 *** |
|  | (0.13) | (24.56) | (5.08) |
| IA | −0.010 * | −0.546 *** | 0.195 ** |
|  | (−1.86) | (−6.29) | (2.09) |
| lngdp | 0.001 ** | 0.054 *** | 0.001 |
|  | (2.05) | (6.86) | (0.13) |
| lntime | 0.001 *** | 0.002 | 0.052 *** |
|  | (3.06) | (0.26) | (10.16) |
| Constant | 0.062 *** | 2.220 *** | −0.798 *** |
|  | (7.23) | (17.62) | (−8.48) |
| Observations | 9955 | 9786 | 9480 |
| R-squared | 0.103 | 0.681 | 0.114 |
| Industry FE | YES | YES | YES |
| year FE | YES | YES | YES |

Robust *t*-statistics in parentheses; ***, ** and * represent significant levels of 1%, 5% and 10%.

The results of the control variables are also generally in line with expectations: the regression coefficients of the ownership concentration ratio (first) reaches a significant level but the sign is negative, indicating that the larger the shareholding ratio of the largest shareholder, the closer the control over the enterprise's operation and profitability, so there may be risk aversion motivations, thereby reducing the enterprise's risk taking level; The size of the board of directors (lnboard) is also negative at the significant level of 1%, indicating that the larger the number of directors, it is often difficult to obtain a unified opinion in decision-making, and there are differences in the willingness to take risks, thereby reducing the level of corporate risk taking; The asset-liability ratio (lev) is significantly positive at a significant level of 1%, indicating that companies with more debt tend to be more willing to take risks in order to get out of business difficulties as soon as possible; The listing age of enterprises (lntime) is significantly positive, indicating that the more mature the enterprise, the stronger the operating strength of the enterprise, and when considering the choice of risk-taking, it often has greater confidence to choose high-yield and high-risk projects; The growth of an enterprise is significantly negative at the 1% significant level, which means that the higher the growth, the more cautious the enterprise is in decision-making, and the lower the enterprise's risk bearing level. Besides, from the external environment, the gross domestic product (lngdp) is significantly positive at a significant level of 5%, indicating that there is a positive impact mechanism between the level of economic development and the level of the corporate risk-taking.

Columns (2) and (3) report the regression results of the impact of the degree of digital transformation on financial risk-taking and strategic risk-taking, respectively. The results show that digital transformation maintains financial stability and increases the level of financial risk-taking (coefficient of 0.01, passing the statistical significance test at 5%), and the regression coefficient between digital transformation and strategic risk taking is significant, and the value is 0.011, indicating that enterprises can increase the level of strategic risk-taking in digital transformation. The reason is that digital transformation may cause certain financial risks, but overall it brings more opportunities to improve the financial risk management of the company. The use of digital technology has created a "digital infrastructure" for reducing and controlling financial risks, which can not only improve the level of enterprise budget management, but also reduce the cost of financial information, and quickly identify potential crises. Moreover, the intelligent platform can promote the efficiency of industrial financial integration, further expand the financial boundary of enterprises, and thus improve the level of financial risk-taking. Furthermore, in the long run,



digital transformation reshapes the enterprise's human resource management, business process, organizational structure, technology and products, promotes the internal financial synergy of the enterprise, optimizes the efficiency of resource allocation, disperses enterprise risks, improves the enterprise's operational efficiency and sustainable development ability, and the enterprise's digital transformation has complied with the requirements of market development, thus improving the enterprise's strategic risks-taking.

### 5.2. Robustness Analysis

(1) Replacement variables: This paper uses the variable replacement method to test the robustness of the previous conclusions. In this paper, the extreme difference (risk2) of the industry-adjusted return on total assets (adj_roa) is used to replace the above corporate risk-taking index. According to the result of column (1) in Table 5, the regression coefficient between the two variables is still significantly positive correlated at the 1% level. The results are consistent with the assumptions above, indicating that the conclusions of this study are reliable.

**Table 5.** robustness tests.

| VARIABLES | (1) Replacement Variables | (2) Excluding Some Samples | (3) Instrumental Variables Method |
|---|---|---|---|
| lnfrequent | 0.002 *** | 0.001 *** | 0.003 *** |
| | (3.74) | (4.73) | (7.18) |
| Constant | 0.325 *** | 0.177 *** | 0.201 *** |
| | (16.83) | (16.85) | (13.51) |
| Under-identification test | | | 0.000 |
| | | | (40.032) |
| Weak instrumental variable test | | | 0.000 |
| | | | (20.13) |
| Over-identification test | | | 0.7212 |
| | | | (0.127) |
| Observations | 10,088 | 9183 | 9325 |
| R-squared | 0.122 | 0.118 | 0.015 |
| Control variables | YES | YES | YES |
| Industry FE | YES | YES | YES |
| year FE | YES | YES | YES |

Robust *t*-statistics in parentheses. *** represents significant levels of 1%. Under-identification tests using the Kleibergen-Paap RK LM statistic, over-identification tests using the Hansen J statistic, and weak instrumental variable tests using the Kleibergen-Paap RK Wald F statistic.

(2) Excluding some samples. Considering that enterprises in the information transmission and software information technology service industries have a high level of informatization [13], compared with other industries, enterprises have fewer obstacles in the process of digital transformation and are not universal. Therefore, this paper eliminates the samples of information transmission and software information technology service industries and conducts regression again. The results in column (2) of Table 5 show that the regression coefficients of Infrequent and risk1 are still significantly positive correlation after deleting some samples, which is consistent with the previous research results.

(3) Instrumental variables approach. To control the endogenous problems and the effects of reverse causality generated arising from omitted variables. the two-stage least squares method is used to perform the regression. The annual industry average of the degree of digital transformation for companies other than our own (average) and keyword word frequency (vcount = total number of keyword words/total number of words in the annual report) are selected as instrumental variables [43]. The test results Table 5 column (3) shows that the regression coefficient is significant and the value is 0.003. At the same time, in order to explain the rationality of tool variable selection, the test results of tool variables are reported in column (3) of Table 5. The results show that: the Wald F-statistic

of 20.13 is greater than 10, which rejects the original hypothesis, indicating that there is no problem of weak instrumental variables; for the under-identification test, the *p*-value of the LM statistic of 0.000 passes the significance test. The value of the Hansen J statistic is 0.7212, which is much larger than 0.1, rejecting the original hypothesis, that is, passing the "over-identification of instrumental variables" test, again indicating the robustness of the paper's conclusions.

### 5.3. Heterogeneity Test

As a large economy in China, there are inevitable differences in attribute characteristics among enterprises. Under the wave of the digital economy, due to the differences between the current business situation and future development direction of enterprises, there may also be differences in digital transformation behavior, and then there is the asymmetry in the effect of the impact on the level of corporate risk-taking. Therefore, it would be helpful to subdivide the sample according to the different attributes of enterprises to avoid generalization of results due to "one size fits all", and to help the government to provide localized policy guidance according to the attributes of different enterprises. Based on the above considerations and the attribute characteristics of listed enterprises, this study will be tested from the property rights attributes and scientific and technological attributes of enterprises, that is, examining the differences in the attributes of "state-owned enterprises and non-state-owned enterprises" and "high-tech enterprises and non-high-tech enterprises". In order to distinguish the samples, this paper sets the dummy variables: State and Tech, where the value of State is 1, indicating a state-owned enterprise, and the value of State is 0, indicating a non-state-owned enterprise; When Tech value is 1, it indicates a high-tech enterprise; when Tech value is 0, it indicates a non-high-tech enterprise.

The results of the empirical analysis are shown in Table 6 below. From the table, we can see that in column (1) and column (2) of the property rights attribute of enterprises, the t-value of the regression coefficient of state-owned enterprises is small and does not pass the significance test, in comparison, the regression coefficient of non-state-owned enterprises passes the significance test of 1%. The possible reasons are that state-owned enterprises are large in scale, large in personnel, diverse in business types involved, clear in the division of responsibilities of various departments within the enterprise, high in professional barriers, and difficult to break the original interest pattern and power system. Therefore, such enterprises need to consider more elements and are subject to greater restrictions in the process of digital transformation and change, and have higher requirements on the overall planning ability and change boldness of decision makers. Moreover, due to the preference of national policies and government support for state-owned enterprises, state-owned enterprises have advantages in obtaining resources and seizing market position, and the competitive pressure for the survival of enterprises is low. At present, state-owned enterprises are not strong enough in their willingness to digital transformation, and are conservative in the formulation of transformation strategic positioning and objectives. Compared with state-owned enterprises, non-state-owned enterprises are facing strong competitive pressure. Under the trend of the digital era, non-state-owned enterprises have a strong desire and motivation for digital transformation in order to seize more market share and a seat in the market. Last but not least, in non-state-owned enterprises, the management and employees are relatively young, the reaction to the market and the application of new technologies are fast, and the resistance to digital transformation is relatively small. Therefore, with the deepening of digital transformation, the ability of enterprises to take risks has also increased significantly. Columns (3) and (4) show the analysis results of different scientific and technological attributes of enterprises. The results show that non-high-tech enterprises and high-tech enterprises show a certain differentiation effect. Although high-tech enterprises passed the 1% significance test, non-high-tech enterprises did not pass the statistical significance test. The possible reason is that enterprise digital transformation is a "full scene", "full intelligence" and "full value" transformation. It is an innovative work and a long-term, continuous trial and error process.

As high-tech enterprises themselves are at the forefront of the scientific and technological era, they have the technical support and talent reserve required for digital transformation. In their own existing business models and business systems, digital models can effectively support business model innovation and cross organizational innovation, so as to give full play to the effectiveness of the digital transformation of enterprises. In contrast, the overall framework awareness of digital transformation of non-high-tech enterprises is weak, and the limitations of traditional automation equipment and the weak foundation of enterprise digital technology support make enterprises face the risk of not being able to turn and not being able to turn. The digitalization level of such enterprises is relatively low, and if enterprises do not consider their own factors and just follow the digitalization trend for forced transformation, they will instead fall into the illusion of transformation and cause greater economic waste. Therefore, the low level of digital enterprises naturally cannot improve the risk-taking level of enterprises.

**Table 6.** Heterogeneity test.

| VARIABLES | (1) Non-State-Owned Enterprises | (2) State-Owned Enterprises | (3) Non-High-Tech Enterprises | (4) High-Tech Enterprise |
|---|---|---|---|---|
| lnfrequent | 0.002 *** | 0.000 | 0.000 | 0.002 *** |
|  | (4.48) | (0.20) | (0.67) | (4.53) |
| Constant | 0.190 *** | 0.162 *** | 0.170 *** | 0.173 *** |
|  | (11.74) | (11.93) | (12.65) | (10.27) |
| Control variables | YES | YES | YES | YES |
| Observations | 6819 | 3271 | 4901 | 5189 |
| R-squared | 0.124 | 0.140 | 0.119 | 0.152 |
| Industry FE | YES | YES | YES | YES |
| year FE | YES | YES | YES | YES |

Robust *t*-statistics in parentheses; ***, represents significant levels of 1%.

## 6. Identification Test of Mechanism Paths

The previous part found that the digital transformation of enterprises will significantly improve enterprise risk-taking, but the relationship between the two has only been described as a whole, and the mechanism black box has not been studied. In this article, the above problems focus on the mechanism of the influence between the two. In this regard, this paper selects three types of channels for verification: innovation-driven, resource allocation efficiency, and value improvement. In order to portray the mechanism path of enterprise digital transformation affecting enterprise risk assumption, this paper sets the following model with the help of the test procedure proposed by Wen Zhonglin and Ye Baojuan (2014) [44]:

$$risk1_{i,t} = \alpha_0 + \beta_1 Infrequent_{i,t} + \sum \varphi CVs_{i,t} + \sum year + \sum Indenty + \varepsilon_{i,t} \tag{6}$$

$$Med_{i,t} = \delta_0 + \theta_1 Infrequent_{i,t} + \sum \varphi CVs_{i,t} + \sum year + \sum Indenty + \varepsilon_{i,t} \tag{7}$$

$$risk1_{i,t} = \alpha_0 + \beta_1 Infrequent_{i,t} \delta Med_{i,t} + \sum \varphi CVs_{i,t} + \sum year + \sum Indenty + \varepsilon_{i,t} \tag{8}$$

where $Med_{i,t}$ is the mediating variables, which are innovation-driven, resource allocation efficiency and enterprise value, respectively. With reference to Lin, Li (2022) [45] and Chen, Xiaohui (2021) [8], the ratio of enterprise R&D investment to business revenue is used to represent enterprise innovation input (RD), and the natural logarithm of enterprise patent applications is expressed as innovation output (lninvent). Drawing on the practice of Lu Xiaojun and Lian Yujun (2012) [46], the LP method is used to calculate the total factor productivity of listed companies (TFP); Referring to Huang, Dayu (2021), Tobin Q is used to measure enterprise value (tbp), which is calculated as Tobin Q = (year-end stock price * number of outstanding shares + net assets per share * a number of non-marketable shares + book value of liabilities) + total assets [47]. Finally, if the coefficient of enterprise digital transformation in model (7) and the coefficient of intermediary variable in model (6) are both significant, it indicates that the intermediary effect exists; At the same time, if the

coefficient of enterprise digital transformation in model (8) is significant, it means that the intermediary variable plays a part of the intermediary effect, if not significant, it means that the intermediary variable plays a full intermediary effect.

### 6.1. Innovation-Driven Effect

In Table 7, based on the perspective of "digital transformation—innovation driven—enterprise risk bearing level", this paper describes the "input-output" performance of enterprise digital transformation on R&D innovation. Column (1) indicates the effect of digital transformation on risk-taking when the mediating variables-innovation inputs and outputs-are not included, while (3) and (5) explore whether the effect of digital transformation on risk-taking changes when the mediating variables are included, so as to demonstrate whether innovation input and output can exist as intermediary variables.

**Table 7.** Result estimation of innovation-driven effects.

|  | (1) | (2) | (3) | (4) | (5) |
|---|---|---|---|---|---|
| **VARIABLES** | **risk1** | **RD** | **risk1** | **lninvent** | **risk1** |
| lnfrequent1 | 0.001 *** | 0.005 *** | 0.001 *** | 0.114 *** | 0.001 *** |
|  | (4.02) | (14.93) | (3.29) | (10.76) | (4.38) |
| RD |  |  | 0.033 ** |  |  |
|  |  |  | (2.57) |  |  |
| lninvent |  |  |  |  | 0.001 *** |
|  |  |  |  |  | (4.52) |
| Constant | 0.178 *** | 0.077 *** | 0.179 *** | −11.876 *** | 0.152 *** |
|  | (17.15) | (7.29) | (15.96) | (−27.70) | (11.96) |
| Observations | 10,090 | 9350 | 9196 | 8659 | 8549 |
| R−squared | 0.124 | 0.415 | 0.117 | 0.376 | 0.114 |
| Control variables | YES | YES | YES | YES | YES |
| Industry FE | YES | YES | YES | YES | YES |
| year FE | YES | YES | YES | YES | YES |

Robust t-statistics in parentheses; *** and ** represent significant levels of 1% and 5%.

The results show that the correlation coefficient in columns (2) and (4) reaches a significant level, indicating that digital transformation has a positive contribution to the R&D input and innovation output of enterprises. The coefficients of the mediating variables in columns (3) and (5) are significant at the 5% and 1% levels, respectively, which indicates that there is an intermediary effect on R&D investment and innovation output, that is, the digital transformation improves the innovation ability of enterprises, thus providing impetus for enterprises to improve the risk-taking level. From the perspective of enterprise R&D investment, enterprise digital transformation is a systematic project, which requires enterprises to have high digital infrastructure, so enterprises need to invest a lot of money to build digital platforms. From the perspective of innovation output, the digital transformation of enterprises makes full use of data resources through digital technology, simplifies the innovation process, reduces the error rate in the innovation process, and improves the efficiency of enterprise innovation output. A company that is good at innovation is bound to have a unique competitive advantage in the market, so the stronger the enterprise's ability to resist pressure, the stronger its willingness to take risks. This is consistent with Hypothesis 3 of this paper.

### 6.2. Resource Allocation Promotion Effect

In Table 8, this paper shifts from the "innovation-driven" perspective to the "resource allocation" study. The coefficient of digital transformation in column (2) is positive and significant, which indicates that digital transformation can promote the efficiency of resource allocation. The coefficient of total factor productivity of the mediating variable in column (3) is negative and significant at the 5% level. The reason may be that through digital transformation, limited resources can be shared or precisely allocated among organi-

zations within the enterprise [48], which reduces the risk of the business process caused by resource distortion or mismatch, so that the risk faced by the enterprise is not so great, and therefore the enterprise does not urgently need to improve its risk management capability. The results of the study show that resource allocation efficiency, as a transmission path, conveys the inhibitory effect of enterprise digital transformation on risk taking.

**Table 8.** Result estimation of the resource allocation promotion effect.

|  | (1) | (2) | (3) |
| --- | --- | --- | --- |
| **VARIABLES** | risk1 | TFP | risk1 |
| lnfrequent | 0.001 *** | 0.000 *** | 0.001 *** |
|  | (4.02) | (8.67) | (3.97) |
| TFP |  |  | −0.155 ** |
|  |  |  | (-2.32) |
| Constant | 0.178 *** | −0.070 *** | 0.160 *** |
|  | (17.15) | (-37.40) | (14.02) |
| Observations | 10,090 | 9741 | 9583 |
| R-squared | 0.124 | 0.733 | 0.118 |
| Control variables | YES | YES | YES |
| Industry FE | YES | YES | YES |
| year FE | YES | YES | YES |

Robust *t*-statistics in parentheses; *** and ** represent significant levels of 1% and 5%.

## 6.3. Value Enhancement Effect

In Table 9, this study uses enterprise value (tbq) as the intermediary variable and risk1 as the proxy variable of enterprise risk bearing level to test the intermediary effect. The results show that the coefficient of enterprise digital transformation in column (2) is 0.029, which is highly significant, and the coefficient of the intermediary variable enterprise value in column (3) is significant, so enterprise value plays a partial mediating effect between digital transformation and enterprises risk-taking. As mentioned earlier, on the one hand, digital transformation greatly reduces the degree of information asymmetry, improves the allocation efficiency and utilization of resources, and thus promotes the increase of enterprise value. On the other hand, enterprises with a high level of digitalization meet the development needs of today's market and are more likely to get the support of external investors, so as to gather more funds to enhance innovation momentum, promote more innovation output, and enhance corporate value. Therefore, the digital transformation of enterprises can promote the improvement of enterprise value, and this positive driving effect will be transferred to the enterprise risk-taking level.

**Table 9.** Result estimation of the value-enhancing effect.

|  | (1) | (2) | (3) |
| --- | --- | --- | --- |
| **VARIABLES** | risk1 | tbq | risk1 |
| lnfrequent | 0.001 *** | 0.029 *** | 0.001 *** |
|  | (4.02) | (3.49) | (3.95) |
| tbq |  |  | 0.001 ** |
|  |  |  | (2.54) |
| Constant | 0.178 *** | 9.945 *** | 0.169 *** |
|  | (17.15) | (26.29) | (16.07) |
| Observations | 10,090 | 10,087 | 9925 |
| R-squared | 0.124 | 0.324 | 0.123 |
| Control Variables | YES | YES | YES |
| Industry FE | YES | YES | YES |
| year FE | YES | YES | YES |

Robust *t*-statistics in parentheses; *** and ** represent significant levels of 1%and 5%.

## 7. Research Findings and Policy Implications

Based on the degree of digital transformation of listed enterprises in 2015–2020, this paper empirically examines the impact of digital transformation on the level of corporate risk-taking and its channel mechanism by using a benchmark model and a mediating effect model, with reference to the estimation results of a static panel model. First of all, this paper judges the impact of digital transformation and the full sample of enterprise risk taking; Secondly, based on the heterogeneity, this paper explores the property right attribute and scientific and technological attribute respectively; Finally, based on the impact path test, this paper examines the total mechanism channel from the three paths: innovation-driven effect, resource allocation promotion effect and value enhancement effect, thus deepening the research on risk-taking of enterprise digital transformation. The results are shown as following:

(1) The digital transformation of enterprises can improve the level of risk-taking. Specifically, digital transformation maintains the stability of financial risks and positively promotes the improvement of financial risk-taking and strategic risk-taking.

(2) According to the regression results of the difference in the characteristics of enterprise attributes, the digital transformation of non-state-owned enterprises and high-tech enterprises can better promote the improvement of the enterprise risk-taking level.

(3) From the perspective of the impact path testing mechanism, the digital transformation of enterprises can improve the R&D input and innovation output of enterprises, enhance the efficiency of resource allocation, and thus promote the increase of enterprise value. Among them, enterprise innovation input-output and enterprise value contribute to the enhancement of enterprise risk-taking level, while resource allocation efficiency as an intermediary path weakens the impact of digital transformation on enterprise risk-taking level.

In general, the digital transformation of enterprises has a significant impact on the level of enterprise risk bearing, which has improved the level of enterprise risk taking. Therefore, enterprises need to seize the opportunity of digital transformation enabled by digital technology, promote the active transformation of enterprises, improve the economic efficiency of enterprises or form new business models, so as to enhance corporate resilience to stress.

According to the above research findings, combined with the degree of digital transformation of Chinese enterprises, risk prevention and governance status, the following policy insights are obtained. First, comply with the development trend of digital economy and implement the application of digital technology. Digital transformation is not accomplished at one stroke, which requires the government to seize the opportunity of developing the digital economy to actively promote the digital transformation of enterprises, give certain policy preferences to encourage enterprises to actively implement digital transformation, provide "green channels" for some enterprises in trouble during the process of digital transformation. At the same time, enterprises actively promote the application of digital technology in all aspects of production and operation, paying attention to the integration and promotion of digital technology and their own traditional business advantages, leading to growth with high quality. Second, clarify the development strategy of enterprises and rationally plan the path of digital transformation. Enterprise digital transformation is a systematic project, involving both the application of technology and the level of organizational change. This requires business managers to plan digital transformation as a long-term development strategy. In the early stage of digital transformation, enterprises need to enhance technological innovation, attach importance to talent training, and promote the transformation of management concepts, so as to gradually clarify strategic goals and clear practice paths, laying a solid foundation for smooth digital transformation. Third, track the whole process of enterprise risk management. Through digital transformation, the innovation level, resource allocation efficiency and enterprise value of enterprises have been greatly improved. However, in the initial stage of transformation, a large amount of capital needs to be invested, and the organizational structure needs to be constantly

reshaped. There are certain risks. Therefore, enterprises should pay close attention to the potential risks brought about by digital transformation, track the sources of risks with the help of digital technology, and propose the best response plan for the enterprise risk management model.

**Author Contributions:** Conceptualization, D.D.; Methodology, D.D.; Formal analysis, M.Z.; Writing—original draft, S.H.; Writing—review & editing, D.D. and S.H.; Project administration, J.X. All authors have read and agreed to the published version of the manuscript.

**Funding:** The major project of the National Social Science Foundation of China "Research on the Construction of China's Market-oriented Green Technology Innovation System" (Grant No. 20&ZD060); The Humanities and Social Science Research Planning Fund of the Ministry of Educa-tion "Empirical Research on the Divergence and Convergence State of the Dynamic Process of Online Learning" (Grant No. 17YJA880014).

**Informed Consent Statement:** Informed consent was obtained from all subjects involved in the study.

**Data Availability Statement:** The dataset generated and analyzed in this study is not publicly available. Dataset is available from the corresponding author on reasonable request.

**Conflicts of Interest:** The authors declare no conflict of interest.

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
