# Peer review of "The Impact Mechanism of Digital Transformation on the Risk-Taking Level of Chinese Listed Companies"

_sustainability, doi:10.3390/su15031938_

Round 1
Reviewer 1 Report
The paper deals with a really new and interesting topic, the relationship between risk-taking and technological transformation. I found the paper well structured, the results well presented, and an econometric part really well applied. Kudos. The authors also offer policy implications and a discussion of their empirical results.
Minor comments.
1. The introduction lacks the research question and the paper's contribution. The authors should expand this section, including and specifying their objective. The contribution and the research question are fundamental elements of a paper's Introduction. Even if the authors then include the contribution in the literature review, I think it is clearer for the reader to understand the objective and contribution of the paper from the outset. Therefore, I suggest amending this section. Furthermore, I suggest that the authors specify the methodology adopted right away, i.e. how they achieve this objective.
2. Please check the Multicollinearity between variables, as the Z-score used as the dependent variable already includes some fundamental (balance sheet) characteristics of the companies.
Otherwise, everything is OK. Really nice paper, well structured and clear.
Typo error
Page 7 row 314 "t"
Author Response
请参阅附件
Please see attachment

Reviewer 2 Report
Thank you for your research which is of interest since it considers two hot topics today: digital transformation and risk management. I have some questions about the content of the paper. At a deep level:
- It is not clear to me the main objective of the research: "positive impact" or "improvement" on enterprise risk-taking, what does it mean? Is it better taking higher risks?? I didn´t find an answer in the paper and I guess not all companies have the same attitude toward risk (and shouldn´t have pending on the sector).
- Research questions are not clear to me. P.e. Hypothesis 1: Digital transformation reduces the stability of corporate finance and has a dampening effect on the level of corporate risk-taking and the level of strategic risk-taking. Aren´t these two different hypothesis (1. Digital transformation reduces the stability of corporate finance 2. Digital transformation has a dampening effect on the level of corporate risk-taking and the level of strategic risk-taking.) The same comment for hypothesis 2. Hypothesis 3, 4 and 5 are not clear either.
- Figure 2. Source? Where is this information coming from?
- Table 6, 7, 8. Which is the difference between column 1 and 3? (always risk1)
From the format point of view:
- page 7 line 302: What is ST, *ST, PT?
- Mistakes in the punctuation makes it difficult to read.
- page 7 line 314 "this paper uses the return on assets (roa) to represent t indicate corporate profitability,". What does this t means?
- References are not presented in order.
General comment: even when the topic is very interesting it is difficult to read and to follow. It is not clearly presented.
Reviewer 3 Report
Well done, a great work.
Author Response
Thank you for your recognition of my dissertation, your approval gives me great encouragement.
Round 2
Reviewer 2 Report
Thank you for considering comments and reviewing the paper in consequence.
I think now it is easier to understand and clearer. I still miss in pg 7 line 310 explaining what ST, * ST or PT status. You explain in the document "response to reviewer" but not in the text. First time you express acronyms you have to say what they mean (p.e. ST (Special Treatment)).